# The Impacts of the Emerging Climate Change on Broccoli (*Brassica oleracea* L. var. *italica* Plenck.) Crop

Anastasios S. Siomos *, Konstantinos Koularmanis and Pavlos Tsouvaltzis

Department of Horticulture, Aristotle University, 54124 Thessaloniki, Greece
* Correspondence: siomos@agro.auth.gr

**Abstract:** Broccoli (*Brassica oleracea* L. var. *italica* Plenck.) is nowadays one of the most important vegetable crops worldwide, with an increasing demand by the market, due to its high nutritional value. Based on the optimal temperature range, its cultivation in the temperate regions takes place from late summer to late spring. Broccoli cultivation nowadays faces many challenges, such as the profitable production throughout the year, as well as during conditions of an increased temperature, induced by the emerging climate change, as well as the possibility of introducing the cultivation to subtropical and tropical areas. The modern genotypes (F1 hybrids) differ significantly among themselves in terms of the requirements for the head's formation, which, however, has not been fully elucidated. In this direction, breeders have been developing programs internationally since the early 1990s, in order to create genotypes that will be adapted to temperatures higher than the optimal range, having, however, a relatively limited initial germplasm pool. The purpose of this review is to present extensively the temperature requirements in the broccoli crop production and to highlight the impacts of the emerging climate change.

**Keywords:** *Brassica oleracea* L. var. *italica* Plenck.; head initiation; head growth; high temperature; heat tolerance; environmental stress



## 1. Introduction

The *Brassica oleracea* L. is a polymorphic species that contains many botanical varieties (var.) with very different phenotypes, with the cole brassicas, e.g., cabbage (var. *capitate* L. f. *alba* DC), Brussels sprouts (var. *gemmifera* DC), broccoli (var. *italica* Plenck), cauliflower (var. *botrytis* L.), kale (var. *acephala* DC) and kohlrabi (var. *gongylodes* L.) being the most important ones worldwide [1,2]. In two of these varieties, such as cauliflower and broccoli, the vegetative growth is terminated at a certain stage of development and the meristem turns into an inflorescence, before the formation of the flower head begins, which consists of the edible part of the plant [3–6]. However, the physiological and genetic basis of these processes are not fully elucidated [4].

Based on FAO statistics [7], in 2020, the global production of broccoli and cauliflower was 25,531,274 tons from an area of 1,357,186 ha, in which Europe ranks third (after China and India) and contributes 9.5 and 10.5% in production and area, respectively. In Mediterranean countries, which are particularly vulnerable to climate change, 57.9% of the cultivated area in Europe is located there, from which 61.8% of European production results. Indicative of the interest that exists worldwide, as well as in the countries of southern Europe, is the fact that in 2020, both the global cultivated area and the production of broccoli and cauliflower increased by 60%, compared to 2000, while during the same period, although the cultivated area and the production in Europe as a whole appeared stable, in southern Europe, the cultivated area increased by 17 and the production by 11% [7].

The increased demand for the product and in periods beyond those with favorable conditions for its production (moderate temperatures), such as during the summer season,

combined with the expected increase in global temperature due to climate change, has created or will create the need for the cultivation of broccoli under heat-stressed conditions. The higher temperatures expected with emergent climate change and the potential for more extreme weather events will affect the productivity of cultivated plants, given that temperature is a primary factor affecting the plant growth rate [6]. The effects of temperature are intensified by the shortage of or an excess of soil water, demonstrating that understanding the interaction between temperature and water will be needed to develop more effective adaptation strategies to offset the effects of higher temperature extremes associated with a changing climate [6].

In this direction, a detailed knowledge of the crop's requirements in the environmental conditions during the various stages of the growth and development and, in particular, the stage of the head's initiation as well as its response to abiotic stresses, with higher than the optimum temperatures being dominant, is considered to be of key importance in terms of the growth, development, yield and quality of the produced broccoli. Therefore, the purpose of this review is to present extensively the environmental requirements in the broccoli crop production (with those dominant in the temperature) and to highlight the impacts of the emerging climate change.

## 2. Plant Growth and Development of Broccoli

From an agronomic point of view, the growth and development of broccoli can be divided into two stages [1,8]. The first one, starting from the transplanting to the onset of the head's formation, includes the meristem differentiation from vegetative to reproductive, given that the appropriate stimulus has been induced [9], and the second one lasts from the onset of the head's formation to its harvest.

The main head is formed at the apex of the central shoot, while secondary heads are formed on lateral shoots that develop from the buds in the leaf axils [10]. A lateral shoot formation and development is not always a desirable trait, thus most of the modern F1 hybrids typically exhibit a strong tip dominance and a limited lateral shoot development [2] until the main head is harvested [11]. Potential transcription factors involved in the lateral eye growth's arrest and restriction have been recognized and identified. However, a strong variability in the lateral shoot's development occurs between genotypes, particularly under stressful environmental conditions [12]. It has been reported that 6.3 and 16.0 lateral shoots/plants have been developed in the commercial varieties of Coastal and Gem, respectively, in the range of 17–22 °C, when only a unique head was formed, while many lateral shoots developed in the range of 17–27 °C, but without simultaneously forming a head [10].

### 2.1. Head Initiation

Head initiation requires the development of a certain minimum number of leaves [1,8, 13–15], half of which are visible to the naked eye at this stage [1,15]. Counting the visible leaves is an easy and non-destructive way to assess the plant growth stage [1,15], given that the total number of leaves is linearly related to the number of visible leaves [15]. It has been reported that 13–31 leaves are required for the head's initiation [13,15,16] depending on the genotype, with the lowest number being required in the early stage and the highest number being required in the late stage.

Therefore, the required number of leaves is primarily related to the genotype [8,13,15,17], while secondarily and partially, it is modified by the temperature in the plant growth environment [1,13–15,18–21].

High temperatures increase the required minimum number of leaves and delay the time of the head's initiation [1,13,15,18–21]. Indicatively, it is reported that the number of leaves visible under the microscope before the formation of the head increased from 16.7 to 26.6, with an increase in the temperature from 13 to 30 °C [21], and from 18 to 24 with an increasing temperature from 12 to 27 °C, while a strong interaction between the temperature and genotype was exhibited [10]. In addition, high temperatures also

increased the plant height. Indicatively, it is reported that the plant height increased from 7.5 to 29.9 cm while the temperature increased in the region from 13 to 30 °C [21]. However, it is worth mentioning that a negative correlation between the final number of leaves and the size of the head has been reported by Lindemann-Zutz et al. [22].

The reports on the effect of the day length (photoperiod) and the intensity of the solar radiation on the plant's growth and development as well as the head's formation are not only limited but are also often contradictory. According to some of these reports, neither the photoperiod [23,24] nor the light intensity and quality [25] had an obvious effect on the head's initiation. In addition, no apparent effect on the time of the head's emergence was observed after reducing the intensity of the solar radiation by 25 or 38%, although the number of leaves was increased by 1–3 [16]. Moreover, under conditions of a different intensity of solar radiation levels up to 70%, the number of leaves did not differ, although the duration of the crop was prolonged [26]. In contrast, according to Gauss and Taylor [21], in plants grown under controlled conditions, the time required from sowing to the head's initiation decreased from 86 to 64 days, while the photoperiod was increased from 8 to 24 h and the light energy consequently increased from 2.4 to 7.25 × 10$^7$ erg/s/cm$^2$, although the number of leaves that had developed before the head initiation was not affected by the photoperiod. Finally, according to Fujime et al. [27], the synergistic effect of the day length with the temperature was observed, but only in some of the commercial varieties. In that case, the longer the day, the higher is the temperature range in which the head's formation is possible.

In order to calculate the required time until the beginning of the head's formation, several thermal time models (accumulated temperature values) have been used, which are based on three basic temperatures: a minimum (base), below which the plant does not grow, an optimum, at which the growth rate is a maximum and a maximum (upper limit), above which the growth rate is zero, although there is no consensus on the proposed values. Values range from 0 to 9.9 °C for basal temperatures, from 10 to 21 °C for optimum temperatures and from 18 to 35 °C for maximum temperatures [8,14,17,20,24,28–33].

For the rate of leaf emergence, a base temperature of 2–6 °C has been proposed, depending on the genotype, while for the optimum temperature there are indications that the value is around 20 °C [9,30]. The rate of the leaf's emergence at 15 °C ranges from 0.2 to 0.4 leaves/day for plants with 4 and 14 visible leaves, respectively [30].

Opinions vary regarding broccoli's requirement to be exposed to low temperatures for the induction of the head's initiation [1,18,21,23,27,33–36]. Indicatively and not exhaustively, it is mentioned that Gauss and Taylor [21], Mourão and Brito [35] and Okazaki et al. [36] consider that broccoli does not require an exposure to low temperatures for the head's initiation. In contrast, Wiebe [23] suggests 0 °C as an inductive temperature and 20 °C as a non-inductive one, with 5 °C being more effective, combined with an exposure time of 2–4 weeks and a minimum number of 4 leaves, in order to accelerate the head's initiation. Hadley and Pearson [9] indicated 14–16 °C as the most effective temperature and Wurr et al. [14] indicated 15.8 °C as the most effective one with 23.6 °C as a maximum, while Grevsen [37] indicated 16.3 °C as the optimum with a base temperature of 2.9 °C and a maximum of 29.7 °C. According to Fellows et al. [20] no head formation was observed at the constant temperatures of 0, 30 and 35 °C, while 96, 51, 36 and 64 days were required for the head's initiation at 5, 10, 15 and 20 °C, respectively. According to Uptmoor et al. [33] an exposure to low temperatures for the head's initiation is required in some genotypes. Finally, according to Fujime and Hirose [18] and Fujime et al. [27], various combinations of the day (10, 15, 20, 25 and 30 °C) and night (10, 15, 20, 25 and 30 °C) temperature or the day length (8 and 16 h) and temperature (17 and 23 °C) prevented, allowed, or accelerated the head's formation.

This discrepancy is apparently due to the different genotypes that have been used, the difficulty in determining precisely the time of the head's initiation, given that there are no known distinct phenological or biochemical changes that can signal this time [30] in the requirement or not for an exposure to low temperatures [33], in possible sensitivity

to a specific (but indeterminate) developmental stage [36,37], as well as in the different experimental approaches applied, in terms of the growth conditions (controlled in growth chambers or diverse in the field). Presumably, an exposure to low temperatures is a facultative requirement [32], meaning that if not essential in all genotypes [36], in some it accelerates the head's initiation [13,27] and clearly these temperatures are higher than those required for vernalization in cabbage [36].

Nevertheless, it is apparent that for the initiation of the head in broccoli, an exposure to temperatures below a maximum non-inductive temperature is required, which has not been precisely determined and apparently varies between genotypes. By some researchers [14,25,38], it is considered that no head formation is observed at temperatures > 23–24 °C and by others [39,40] that at temperatures > 30 °C (almost the optimal temperature for vegetative growth), many problems are caused in the head's formation. According to Grevsen [15] and Wurr et al. [14], at mean daily temperatures > 20 °C, late and abnormal head development was observed in the commercial cultivar Shogun, but not in the commercial cultivars Caravel and Emperor. At higher than the optimum temperatures, depending on the value and duration, plant growth stage and genotype, even if the head is formed, characteristic lesions appear on the subsequent head [39]. The stage immediately after the differentiation of the meristem from vegetative to floral and up to the head size of 5–10 mm is considered to be the most sensitive [39,40]. This sensitive stage can be accurately identified microscopically [40], however empirically, it is placed at about 3 weeks before the head is harvested [39].

*FLOWERING LOCUS C* (*FLC*) homologues in *Brassica* species are considered responsible for controlling the requirement for an exposure to low temperatures (vernalization) [41,42]. Although five of them have been cloned and partially or fully characterized in *Brassica oleracea* species (*Brassica oleracea FLOWERING LOCUS C, BoFLC1, BoFLC2, BoFLC3, BoFLC4* and *BoFLC5*) [36,41–43], it is not ascertained which one or which of them are decisive [36]. However, a sensitivity to high temperatures during the initiation stage of the broccoli's head formation is thought not to be associated with genes controlling the vegetative-to-floral meristem differentiation [40], but with one or more genes controlling the growth processes of the floral buds that make up the head [40,44–46]. Given that at this stage the genes *BoiAP3* (*Brassica oleracea itaIica APETALA3*) and *BoiCAL* (*Brassica oleracea itaIica CAULIFLOWER*) are expressed [3], it is possible [40] that their weak or delayed expression at high temperatures is related to the head characteristics due to heat stress that degrades its quality. Recently, the *BoFLC3* (*Brassica oleracea FLOWERING LOCUS C3*) gene has also been proposed [47] as a candidate for the control of the head's initiation under subtropical conditions (high temperature and short-day length). Finally, studies on gene expression in genotypes, that are capable of forming the head under heat stress conditions, suggest the possible involvement of compounds and hormones (such as thioglycosides and jasmonic acid as well as auxin, cytokinin and abscisic acid) in the head's formation under high temperature conditions [47].

The first stage of the growth and development (from the transplantation to the head's initiation) and up to a head growth of 0.6 mm in diameter is the most unpredictable period [8,37], which suggests the involvement of other factors besides temperature [8,33]. The time required for this stage varies more than the time for the second stage (from the head's initiation to the harvest) in early genotypes, while the reversed is observed in the late ones [13]. The rate of the leaf's emergence is mainly a plant developmental process, while the head's initiation is mainly a meristem differentiation process [30], and these two have different optimum temperature regimes. The observations under the field conditions [22,48] demonstrated a strong variability in the time of the onset of the head's formation in the plants of a broccoli cultivar. This is the reason for the great variability in the time of the harvesting of the heads as well, with the consequence that a considerable period of time elapses from the start of the harvest in a crop and up to its completion. The reason for this variability remains unknown. In this direction, no effect of the size of the

seedling at the establishment of the culture was found on the time of the head's initiation nor the growth rate of the head and the final size of the plant [22].

Many mathematical models have been proposed to determine the starting time of the head's formation, among which the following are only indicative and not exhaustive.

According to Mourão and Brito [35], the required thermal time ($\theta$, degree days) for the head's initiation is calculated from the equation:

$$\theta = \sum_{j}^{n}(T - Tb)$$

where $\Sigma$ is the sum, $n$ is the time (days), $T$ is the mean daily temperature (°C) and $Tb$ is the base temperature of 0 °C. Indicatively, it has been calculated that the required thermal time for the initiations of the head in four commercial varieties (Compacta, Comanche, Green Valient and Marathon) amounted to 680 degree days.

According to Grevsen [37], the effective sum of the temperature required for the head's initiation in the commercial cultivars Emperor, Caravel and Shogun is estimated to be 132 degree days. This means that, theoretically, at the optimum temperature, the plants will reach this stage in about 15–18 days from transplanting the seedlings with 3–4 true leaves.

According to Uptmoor et al. [33], the time until the head's formation is divided into the juvenile phase, the vernalization phase and the vegetative-to-reproductive transition phase. The juvenile phase is considered to be independent of the genotype and is completed at a temperature sum of 456 degree days when the plants have approximately four true leaves. Commercial broccoli cultivars are believed to be insensitive to temperature before this stage [23], in contrast to the vernalization phase [32] in which high temperatures increase the time required for the head's formation. The thermal time required for the head's formation was described separately for each genotype using linear regressions with a base temperature of 6 °C and a constant, which has been derived from experiments in growth chambers at 6, 12 and 18 °C [32,33].

To determine the time required from the crop's establishment to the beginning of the head's formation ($1/f$, days) in relation to the average daily air temperature ($T$, °C), the following linear regression has been proposed:

$$1/f = -1.02 \times 10^{-3} + 1.48 \times 10^{-3}\,T$$

Indicatively, for an average daily air temperature of 9.0 and 21.5 °C, the required time is 81 and 32 days, respectively [35].

Fujime and Okuda [34], after 7 years of experimentation in field conditions, propose the following mathematical equations to determine the required time (days) for the head initiation in the commercial variety Wase-midori:

$$Y = 119.071 - 0.409X_1 - 3.543X_2 - 2.232X_3 + 0.297X_4 - 1.397X_5,$$

where:

$X_1$ = the average temperature of the period of 10 days from planting;
$X_2$ = the maximum temperature of the period of 20 days from planting;
$X_3$ = the maximum temperature of the period of 30 days from planting;
$X_4$ = the average temperature of the period of 30 days from planting;
$X_5$ = the maximum temperature of the period of 40 days from planting.

Based on this relationship, the time required during ten crops in different years or periods of the year was calculated to be 29–39 days from planting, these values being in line with the actual ones. In contrast, Gauss and Taylor [21] observed that the time required from the sowing to the head's initiation in plants grown under controlled conditions at a constant temperature of 13 or 29 °C did not differ and were 75 and 74 days, respectively.

Finally, according to Uptmoor et al. [33], the time required from the establishment of the crop of a particular commercial variety to the onset of the head's formation can be easily and accurately determined, while the genotype × environment interaction is also predictable to some extent, using a specific mathematical model only when the conditions are favorable for the formation of the head. On the contrary, the accuracy of the determination is significantly reduced when high temperatures prevail.

Failure to form a head has been reported in cases where there is a failure of apical meristem development (blindness). This is believed [49,50] to be related to either the cessation of the leaf's formation from the apical meristem or a failure to differentiate the vegetative to the reproductive meristem and is caused by unspecified soil-climatic factors, while a significant variation between the genotypes has also been reported.

### 2.2. Head Growth

During the second stage (from the head's formation to the harvest), the head's growth and development is primarily affected by the temperature, as well as the solar radiation, particularly at high plant densities [15]. Thus, simple linear or complex mathematical models have been proposed to determine the head growth rate based on the temperature alone [1,10,13–15,34,35,37,51–54] or solar radiation as well [1,15,21,48,51,52,54].

These mathematical simulation relationships make it possible to predict the required time from the transplanting to the harvest, based on the climatic conditions of an area, in order to schedule the crop planning. In addition, some of them are able to estimate the time of the initiation of the head's formation in order to detect early enough any possible problems in the growth and development of the head, under the prevailing environmental conditions upon the establishment of a crop.

Regarding the temperature levels and, in particular, the minimum and maximum, the sum of the temperatures, the sum of the degree days or active degree days, with or without a base and upper limit temperatures, has been used. As a base temperature, below which the growth and development stop, temperatures of 0, 0.6 and 7 °C have been suggested for various commercial varieties [13,15,37,51] and as an upper limit temperature, that of 17 °C [15,37]. Fujime and Okuda [34], after 7 years of experimentation in field conditions, proposed a mathematical relationship to determine the required time (days) to harvest that is proportional to the time required to the head's initiation, based on the mean and maximum air temperature of various periods since planting. Based on this equation, the time required during ten crop cycles of a genotype in different years or periods of the year was calculated to be 44–71 days from planting, these values being in accordance with the actual ones. According to Grevsen [15], the total temperature requirement of the commercial cultivar Shogun to reach harvest (12 cm head diameter) is 600 degree days.

Regarding solar radiation, the cumulative solar radiation ($MJ/m^2$) index has been used [15].

Therefore, the two stages of the growth and development of broccoli in the field are strongly dependent on the environmental conditions, but each responds differently to them, which in turn makes it difficult to predict the effects of environmental variation [9,37].

Changes in the environmental conditions (temperature, solar radiation, relative humidity, etc.) during cultivation significantly affect both the growth and development stage of broccoli and therefore the quality and yield, while they are often causing difficulties in the production, which differs between genotypes. Indicatively, it has been reported that high temperatures (daily mean > 20 °C) caused a late and abnormal head development in the commercial cultivar Shogun, while the growth of the commercial cultivars Caravel and Emperor was not affected [15]. Additionally, in an experiment under controlled environmental conditions, temperatures > 20 °C during the first month after transplanting inhibited or greatly delayed the head initiation in the commercial cultivar Shogun [14].

However, little information is available on the effect of environmental conditions on the plant's growth and development under field conditions, given that most of the available information comes from an experimentation under controlled conditions and the

application of models derived from these conditions. This issue is also complicated by the fact that plants are at different stages of growth at the time of transplanting, and other, as yet undetermined, factors are likely to be involved [30].

## 3. Climate Change

Climate change is a particularly complex phenomenon and refers to global warming, which is basically due to the increase in the concentration of atmospheric $CO_2$, given the almost linear relationship between them. According to the Intergovernmental Panel on Climate Change (IPCC), the global surface temperature has increased by 0.99 (0.84–1.10) °C from the period 1850–1900 until the first two decades of the 21st century (2001–2020) [55].

The increase in temperature is expected to continue, while at the same time changes are expected also in the amount and distribution of precipitation, as well as in wind patterns, that will lead to a rise in extreme weather phenomena (drought, floods, hailstorms, etc.) which will induce salinity problems as well [56,57].

Therefore, anticipated climate change is related to the variability of four factors (temperature, drought, salinity and $CO_2$ concentration), with the first three causing environmental stresses on plants and leading to crop production losses. Most of the vegetable crops are more sensitive to extreme climatic conditions, such as high temperatures and a limited soil moisture [58], conditions that are also the cause of reduced yields and degraded quality.

These changes in the environmental parameters, that induce stress in plants, are characterized by a strong variation between regions, both on the planet and within countries, with the Mediterranean basin being the most sensitive and therefore more vulnerable to climate change, as the average temperature of the region has already increased by 1.3 °C, in comparison to the temperature levels of the period 1880–1920, while the average global temperature increase was 0.85 °C [56].

As for Mediterranean countries, an indicative reference is made only to Greece. The Committee for the Study of the Effects of Climate Change (CSECC) in its relevant report in June 2011 for the Greek territory [59] estimates that at the end of the 21st century, there will be an increase in the air temperature by 3.1–4.5 °C, based on different climate scenarios, in comparison to the reference decade of 1991–2000 (Table 1). The increase in temperature is estimated to be greater in continental areas, as well as in summer and autumn. This anticipated increase in temperature will cause an increase in the number of days with temperatures > 35 °C by 35–40, especially in the mainland of Northern Greece (where broccoli is mainly cultivated).

**Table 1.** Mean values of air temperature and relative humidity at 2 m from the surface, precipitation, incoming at the surface total short wavelength radiation and the cloud cover fraction for the time periods 1961–1990 *, 2071–2080, 2081–2090 and 2091–2100, as well as the changes in these parameters for the periods 2071–2080, 2081–2090 and 2091–2100 in relation to the reference period 1961–1990. The results are given as the mean and standard deviation of 13 simulations for the A2 emission scenario and 8 simulations for the B2 emission scenario, respectively, and are based on the Prudence program. A2 emission scenario: rapid increase in the atmospheric $CO_2$ concentration, which will reach 850 ppm by 2100. B2 emission scenario: increase in the atmospheric $CO_2$ concentration with moderate but steady rates, which will reach 620 ppm by 2100 [59].

| | Mean Values | | Change (Δ) | | Change (%) | |
|---|---|---|---|---|---|---|
| | Emission Scenarios | | Emission Scenarios | | Emission Scenarios | |
| Periods | A2 | B2 | A2 | B2 | A2 | B2 |
| | Air temperature 2 m above the ground (°C) | | | | | |
| 1961–1990 | 16.17 ± 0.68 | 16.14 ± 0.56 | | | | |
| 2071–2080 | 19.58 ± 0.80 | 18.81 ± 0.67 | 3.41 ± 0.42 | 2.66 ± 0.19 | 21.1 ± 2.8 | 16.5 ± 1.0 |
| 2081–2090 | 19.93 ± 0.82 | 18.94 ± 0.71 | 3.76 ± 0.49 | 2.80 ± 0.34 | 23.3 ± 3.2 | 17.3 ± 2.1 |
| 2091–2100 | 20.64 ± 0.80 | 19.25 ± 0.72 | 4.46 ± 0.38 | 3.11 ± 0.39 | 27.6 ± 2.6 | 19.3 ± 2.5 |

**Table 1.** *Cont.*

| Periods | Mean Values | | Change (Δ) | | Change (%) | |
|---|---|---|---|---|---|---|
| | Emission Scenarios | | Emission Scenarios | | Emission Scenarios | |
| | A2 | B2 | A2 | B2 | A2 | B2 |
| | Air relative humidity 2 m above the ground (%) | | | | | |
| 1961–1990 | 68.47 ± 4.27 | 69.49 ± 4.63 | | | | |
| 2071–2080 | 66.45 ± 2.99 | 68.42 ± 5.02 | −2.02 ± 2.28 | −1.07 ± 0.79 | −2.8 ± 2.9 | −1.6 ± 1.2 |
| 2081–2090 | 65.50 ± 3.04 | 68.14 ± 5.03 | −2.97 ± 2.20 | −1.35 ± 0.72 | −4.2 ± 2.7 | −2.0 ± 1.0 |
| 2091–2100 | 65.23 ± 2.99 | 68.68 ± 4.80 | −3.24 ± 2.09 | −0.81 ± 0.76 | −4.6 ± 2.6 | −1.2 ± 1.1 |
| | Precipitation (mm/year) | | | | | |
| 1961–1990 | 510.1 ± 108.0 | 524.1 ± 113.8 | | | | |
| 2071–2080 | 442.7 ± 112.9 | 497.4 ± 108.6 | −67.4 ± 34,6 | −26.7 ± 50.2 | −13.8 ± 7.6 | −4.6 ± 9.8 |
| 2081–2090 | 397.1 ± 99.6 | 475.7 ± 109.0 | −113.0 ± 29.5 | −48.4 ± 36.4 | −22.6 ± 5.5 | −9.2 ± 8.2 |
| 2091–2100 | 437.7 ± 126.6 | 525.2 ± 138.0 | −72.4 ± 51.1 | 1.1 ± 54.5 | −15.2 ± 10.9 | −0.4 ± 11.2 |
| | Incoming at the surface total short wavelength radiation (W/m$^2$) | | | | | |
| 1961–1990 | 196.1 ± 20.8 | 203.0 ± 21.9 | | | | |
| 2071–2080 | 199.0 ± 19.9 | 206.0 ± 18.3 | 2.9 ± 4.2 | 3.0 ± 5.3 | 1.6 ± 2.5 | 1.7 ± 3.2 |
| 2081–2090 | 201.0 ± 19.6 | 207.2 ± 18.5 | 4.9 ± 4.9 | 4.2 ± 4.8 | 2.6 ± 3.1 | 2.3 ± 3.0 |
| 2091–2100 | 200.5 ± 20.0 | 205.2 ± 18.4 | 4.5 ± 5.4 | 2.3 ± 5.2 | 2.4 ± 3.3 | 1.4 ± 3.1 |
| | Cloud cover fraction (%) | | | | | |
| 1961–1990 | 35.8 ± 4.4 | 36.4 ± 2.1 | | | | |
| 2071–2080 | 31.7 ± 4.3 | 33.3 ± 3.1 | −4.0 ± 1.6 | −3.1 ± 1.4 | −11.3 ± 4.3 | −8.8 ± 4.2 |
| 2081–2090 | 30.2 ± 4.2 | 32.7 ± 2.6 | −5.5 ± 1.7 | −3.8 ± 1.0 | −15.5 ± 4.3 | −10.4 ± 2.9 |
| 2091–2100 | 30.0 ± 4.1 | 33.6 ± 3.1 | −5.7 ± 1.8 | −2.9 ± 1.7 | −16.1 ± 4.8 | −8.0 ± 4.8 |

* Small differences in the estimates of climate parameters in the 1961–1990 reference period for the different emission scenarios are due to the fact that the climate parameters are estimated from different sets of climate simulations for the different scenarios.

## 4. The Impacts of Emerging Climate Change on Broccoli Crop

Climate change is a global natural threat to all sectors of human activity (such as trade, industry, tourism, etc.) but especially to the agricultural sector, given that it is highly dependent on the weather and climate, making the sector an adventurous economic activity [57,60]. Many reviews have revealed that warming has a strong impact on crop productivity [58,61–66], as well as in the produce's quality and nutritional value [66–70].

The rate of the plant's growth and development depends on the ambient temperature and for each species there is a specific range of values that represents a minimum and a maximum value, within which there is a narrower range of values that is characterized as optimal. Responses to temperature differ between both species and the developmental stages of a species [6].

Broccoli is a crop that thrives in periods with moderate (5–25 °C) temperatures [6], with optimal values in the range of 15–23 °C, while a significant degradation of the head's quality is caused under thermal conditions stress (temperatures > 25 °C), with exact values dependent on the genotype [24,39,40,43,71–74]. Determining accurately the temperature thresholds for each genotype as well as the critical stage are very important aspects of assessing the risks from the emerging climate change and exploring the appropriate adaptation options [75].

Recent results [76] showed that higher than normal temperatures affected all the parameters of broccoli plants grown under field conditions (such as the plant height, plant leaf number and weight, stem length and weight, head diameter, weight and quality), except the number of lateral shoots. In summary, growing three F1 hybrids (Cigno, Principe and Domino) in periods with higher (by 4.7–4.8 °C) than normal temperatures resulted in an increase in the plant's leaf number, a decrease in the head diameter, an increased amount of water for the irrigation and extended period from the beginning to the end of the harvest in all three genotypes (Figure 1), while, in addition, it modified the head weight and quality, marketable head weight, marketable crop yield, non-marketable production, loss of leaves and the number of days from planting to harvest (Figure 1 and Table 2), but

in a diverse way in the three genotypes. Higher than normal temperatures caused a wide range of changes in the head's characteristics that degrade its quality (Figure 2).

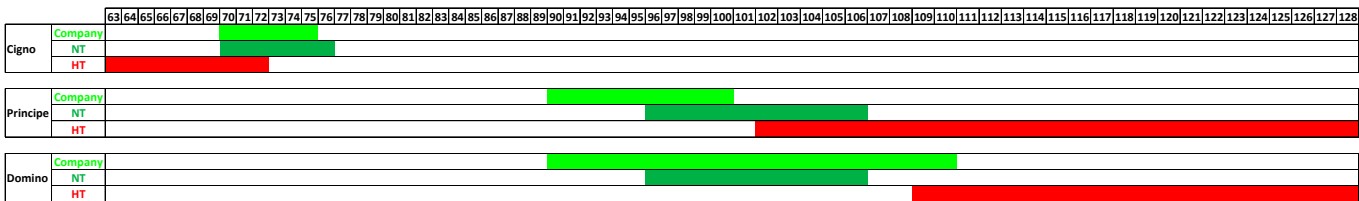

**Figure 1.** The number of days from crop establishment to start and end of harvest of three F1 broccoli hybrids (Cigno, Principe and Domino) based on their seed handling company (Company) and their field cultivation under normal conditions (NT) and of high temperatures (HT) at the Farm of the University of Thessaloniki. The cultivations were established on 7 June (HT) and 4 August 2021 (NT). The average air temperature was 27.8 ± 0.07 and 23.4 ± 0.55 °C in the first (HT) and second (NT) periods, respectively, for the F1 Cigno hybrid, 26.2 ± 1.08 and 20.9 ± 0.57 °C for the F1 Principe hybrid and 26.3 ± 1.18 and 20.9 ± 0.57 °C for the F1 Domino hybrid.

**Table 2.** High temperatures cause a wide range of changes in the plant morphology, cultivation, as well as the yield and quality of the crop.

| Impact Groups | Specific Impacts |
|---|---|
| Plant growth and development | Most parameters (plant height, plant leaf number and weight, stem length and weight) except of the number of lateral shoots<br>The rate of plant growth |
| Head | Initiation of head formation<br>Diameter<br>Weight<br>Characteristics |
| Quality of the produce | Presence of leaves between flower buds<br>Uneven flower bud size<br>Uneven head surface<br>Undesirable head coloration |
| Crop productivity | Marketable crop yield<br>Non-marketable production |
| Cultivation | Days from planting to harvest<br>Duration of the harvesting period |

The changes in the head's characteristics due to the high temperatures are described as an uneven flower bud size, the presence of leaves between the flower buds, an uneven head surface, a reduced head weight and diameter, an undesirable head coloration, etc. [39,40,45, 74,77–79], while above a certain temperature threshold, no head is formed [79]. However, these effects are related to the temperature, plant growth stage and genotype [39,40,74].

The head's quality is the resultant of a set of up to 10 characteristics, while its assessment is difficult and largely subjective [80]. However, defects in the bud's morphology, such as an uneven flower bud size [76,81] and the presence of leaves on the head [76,77] contributes to the degradation of the head's quality, often making it unmarketable.

Bud size uniformity requires that older buds stop growing until younger ones reach the equivalent size, requiring a complex coordination [12]. Lin et al. [47] identified a quantitative trait locus (QTL), a region of DNA associated with the reduction in unequal buds, and a gene (*PERIANTHIA, PAN*) as the most likely candidate within this region. However, it is worth mentioning that by comparing the QTLs maps constructed from three different F2 populations, a total of 86 QTLs controlling 8 traits related to the head's formation in *Brassica oleracea* were identified [44].

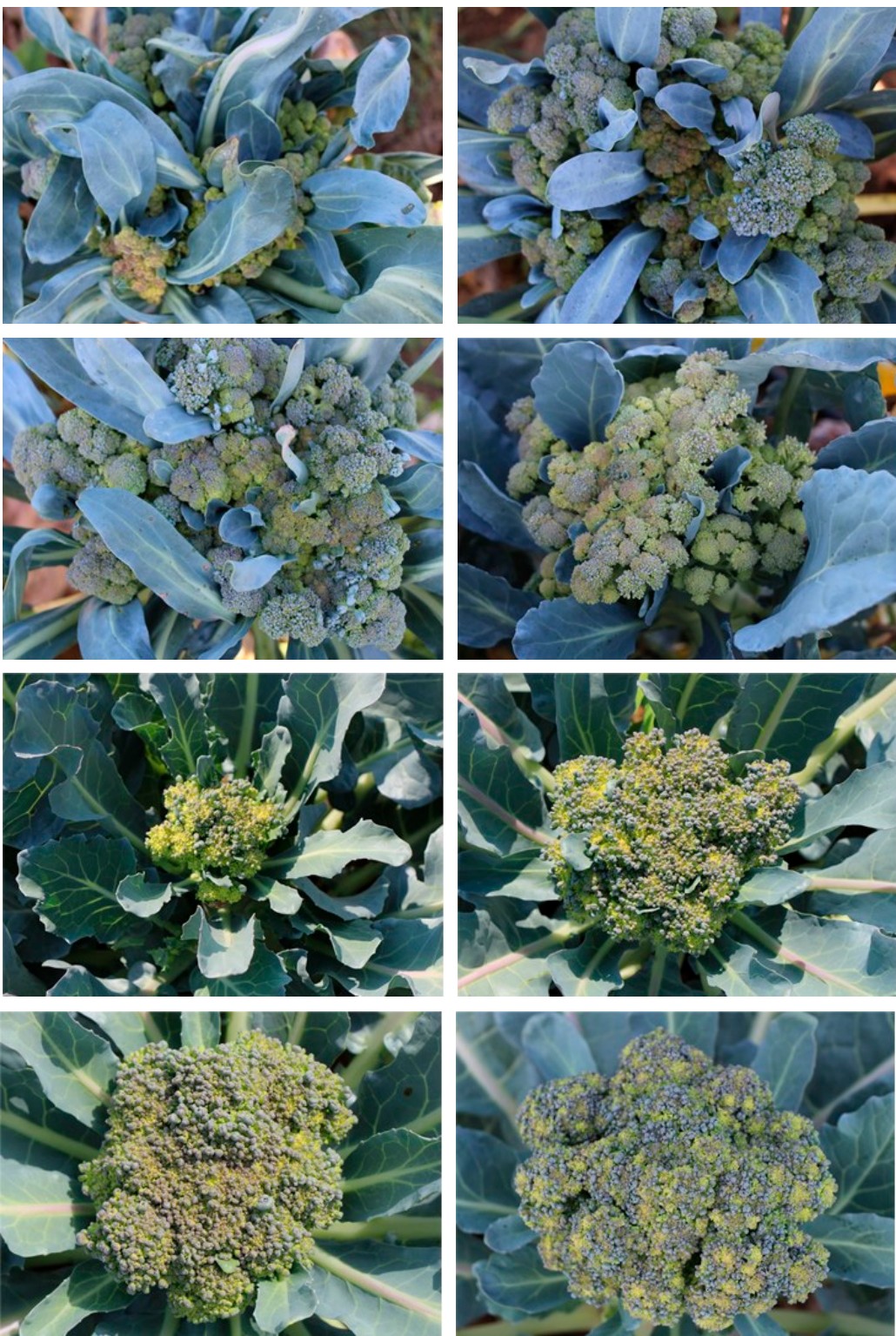

**Figure 2.** Some of the changes in broccoli head's characteristics due to heat stress that degrade its quality: uneven flower bud size, presence of leaves between the flower buds, as well as uneven surface, undesirable coloration and smaller head diameter and weight. The above was observed during the cultivation of three F1 hybrids (Cigno, Principe and Domino) at the Farm of the University of Thessaloniki in the period 7 June–18 August 2021 (Cigno, absolute minimum, average and absolute maximum air temperature 15.3, 27.8 and 41.7 °C, respectively) and 7 June–13 October 2021 (Principe and Domino, absolute minimum, average and absolute maximum air temperature 10.7, 26.2 and 41.7 °C, respectively).

The presence of leaves on the head, that is a common problem in commercial cultivars sensitive to high temperatures [81], is controlled by a single dominant gene and can be suppressed by some alleles of *Brassica oleracea AP1a* (*BoAP1a*) [77], although Stansell et al. [12] suggested that additional candidate genes may be involved. However, the presence of leaves on the head is also influenced by the environmental conditions, with plants grown at higher than optimal growth temperatures (>22 °C) being the most susceptible [28].

Although there has been considerable research investigating the process of the head's formation under both optimal and heat stress conditions [4,31,36,82], the exact genetic basis of this process remains unknown [5,31,34,40,44,83,84], resulting in a lack of information about the complex mechanisms governing the response of broccoli to the heat stress conditions [74].

From the relevant research effort [37,38,85,86], no markers closely related to the heat stress tolerance have emerged, nor have possible candidate genes been proposed, based on publicly available broccoli genome annotation [87]. Although researchers started from the late 1980s to construct genetic maps, to detect the loci of important traits, such as the disease resistance, head morphology, nutritional components, time of the head's formation and maturation, etc. [88–94], these early maps are difficult to unify [85,92,95], due, among others, to the lack of reference genomes for *Brassica oleracea* L. prior to 2014 [86].

However, it has been suggested [74] that a tolerance to heat stress may be the result of the contribution of multiple responsible genes, which are related to the corresponding characteristics of the head, with the most likely candidates being those that encode proteins involved in flower development. Today, it is accepted [10] that the improvement of broccoli and other botanical groups or cultivars of *Brassica oleracea* L. species is limited by complex multi-gene interactions affecting the plant's architecture, developmental processes and yield.

Nevertheless, several broccoli genotypes with a relative resistance to heat stress have been developed in the last two decades [9,79,82,96–100] and some of them possess the ability to grow and produce efficiently in high temperature conditions (average daily temperature 30–36 °C throughout the growing season), under which the commercial varieties Arcadia, Greenbelt, Marathon and Patron fail, as long as the plants constantly remain in the vegetative stage and do not form marketable heads [82]. During their evaluation, however, on a commercial scale, a significant interaction of the soil-climatic conditions in the growing environment was observed [82], with the consequence being that studying the interaction of the genotype within the growing environments has become increasingly important in recent years.

## 5. Challenges and Future Perspectives

Broccoli is today one of the most important vegetable crops worldwide, with an increasing demand for the product from the markets, due to its high nutritional value and especially the high content of secondary metabolites [2,93]. Out of these metabolites (thioglycosides), glucoraphanin has attracted the most intense research interest, given that from its hydrolytic breakdown by the action of the myrosinase enzyme, sulforaphane is produced, which is a compound with a chemoprotective effect against certain forms of cancer [2,101–115].

The botanical varieties or groups in the *Brassica oleracea* L. species (cabbage, Brussels sprouts, broccoli, cauliflower, kale and turnip) are crops with an optimal temperature range of 15–18 °C for vegetative growth [73], while during the initiation of the head's formation in broccoli, temperatures in the range of 15–23 °C [79] or an average temperature < 25 °C [11] are considered optimal. Thus, in the temperate regions, their crop cycle is from autumn to spring.

Due to the increasing demand from the market, the cultivation of broccoli today faces many challenges, such as the possibility of production throughout the year, but also in conditions of an increased temperature, due to the anticipated climate change, in the

temperate regions, as well as the possibility of introducing the crop into subtropical and tropical regions [11,93].

Today, it is widely accepted [81] that modern genotypes (F1 hybrids), the result of intensive breeding efforts targeting mainly to a high yield, a high efficiency of the use of nutrients, the accumulation of the bioactive components, a resistance to pests and diseases and the duration from the crop's establishment to the harvest [2,80,93,116] differs significantly with each other in their requirements for the initiation of the processes leading to the head's formation [2,117–120], which, however, has not been fully elucidated.

In this direction, breeders have been working internationally since the early 1990s [78], aiming to create genotypes adapted to temperatures higher than the optimal range. Initially, all the available commercial varieties were evaluated in summer crop trials. Although more than 60 commercial varieties were examined, only 5 exhibited a potential value for satisfying the objective. Therefore, it is given that breeding programs for the adaptation to high temperatures are based on a relatively small pool of germplasm. Furthermore, without a prior knowledge of how high a temperature tolerance can be inherited in broccoli, the traits were assumed to be polygenic with complex and easily misinterpreted phenotypes and for these reasons, progress is limited. However, before starting any breeding effort, a critical question remains for the breeders [78]: what is the detrimental effect of abiotic stress (high temperature) on the crop in order for it to be overcome?

In this regard, the study of the interaction of the genotype with the growing environment is becoming increasingly important [121]. Comparing genotypes in specific environments is a common approach to evaluating the performance and identifying the traits associated with an improved performance. The analysis of this interaction is considered fundamental in the Mediterranean regions, as they are characterized by a high interannual variation of the climatic factors, and thus of the yield. However, the relevant literature references are extremely limited [122].

The effects of the average temperature on marketable and nonmarketable production are difficult to interpret. In this direction, the temperatures that prevailed during the stage immediately after the differentiation of the meristem from vegetative to floral and up to the size of the head at 5–10 mm might be of a greater interest than the average temperature, as long as this stage is considered to be the most sensitive to temperature [39,40].

However, this stage is difficult to be precisely determined during the plant development [40], while observations in the field conditions [22,48] also showed a strong variability between the plants within a crop, due to the non-simultaneous growth and development, while other unspecified factors are likely to be involved [30]. On the other hand, the temperature values that exert an adverse effect on this stage have not been precisely determined and obviously vary between genotypes.

Furthermore, the current assessments of the impact of climate change have focused on changes in the mean air temperature, which is shaped by the maximum and minimum. It is obvious that the maximum temperature in the plant's microenvironment is different from the one determined by the weather stations [6], given that this is measured in on-site local conditions and especially the soil moisture content and heat loss, due to the water evaporation [123], while the minimum air temperature may be more important in terms of its effect on the growth and phenology, given that it affects the rate of plant respiration at night and can potentially affect the crop's growth, development and yield [124].

Apart from these, there are also limited reports on the effect of solar radiation intensity on the growth and development of the plant and the formation of the head, while the results are also contradictory. On the contrary, the effect of the average temperature on the plant's morphology was undisputed and this has been well documented by many previous studies [1,10,12–15,18–21,124–126], as well as in the amount of water required by irrigation.

It is obvious that the management of the broccoli crop under these conditions is difficult, and the effects of these conditions are particularly adverse. For this reason, the understanding (through description, recording and measurement) of genetically driven growth processes and their correlation with the environmental factors, such as the temper-

ature, radiation, photoperiod, nutrient availability, etc., will contribute decisively to this direction and will help develop adaptation strategies to offset these impacts [1].

## 6. Conclusions

Broccoli is a cool-weather crop, with the head as the edible part, whose formation processes are strongly temperature dependent. It is apparent that temperatures below a non-inducible maximum are required for the head's formation, which has not been precisely determined and apparently varies between genotypes. The stage immediately after the differentiation of the meristem from vegetative to floral and up to the size of the head at 5–10 mm is considered the most sensitive, while at the same time, this period is also the most unpredictable for the crop. This indicates the involvement of other factors besides the temperature (such as compounds, hormones, etc.), with the consequence being that the mechanisms governing the response of broccoli to heat stress conditions are not known and this makes it difficult to predict the effects of any environmental variations. However, it is indisputable that high temperatures (with the exact values depending on the genotype) cause a wide range of changes in the plant both during the stage from the transplanting to the initiation of the head's formation, and from the initiation of the head's formation to its harvest, which affects the duration of the crop, the required cultivation practices, the required time to start the harvest and its duration, as well as the quality and yield of the crop. It is expected that managing the broccoli crop under these conditions will be challenging, given that the schedule of the crop production is expected to change (with a mandatory shift towards a delayed establishment date), as well as the duration of the crop (by shortening or extending it, depending on the genotype), the cultivation practices (a lower plant density, due to the greater vegetative growth, increased requirements in the irrigation water, due to a higher evapotranspiration, a modification of the fertilization rates, due to a greater vegetative growth and a diverse cultivation period, etc.), the duration of the harvesting period and others, which are currently difficult to estimate.

**Author Contributions:** Conceptualization, A.S.S.; writing—original draft, A.S.S., P.T. and K.K.; review and editing, A.S.S. and P.T.; photos were taken by A.S.S. All authors have read and agreed to the published version of the manuscript.

**Funding:** This research received no external funding.

**Data Availability Statement:** Not applicable.

**Acknowledgments:** The authors greatly thank Theologos Koufakis for a helpful discussion and critical reading of the manuscript and Dimitrios Gerasopoulos for a critical reading of the revised manuscript.

**Conflicts of Interest:** The authors declare no conflict of interest.

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
