# Peer review of "The Impacts of the Emerging Climate Change on Broccoli (Brassica oleracea L. var. italica Plenck.) Crop"

_horticulturae, doi:10.3390/horticulturae8111032_

Round 1

Reviewer 1 Report

The current manuscript entitled “The impacts of the emerging climate change on broccoli crop” by Siomos et al. deals with the temperature requirements in broccoli crop production and highlights the impacts of the emerging climate change. After a careful review, I found this manuscript interesting and suitable for publication in the Horticulturae journal. However, I have several comments on the currently submitted version of the manuscript. It needs a thorough major revision. My specific comments are:

1.      Include the scientific name and botanical authority of broccoli in the title.

2.      Production or cultivation? Authors should be consistent while writing these terms. Both terms have differences.

3.      Line 14: breeders of which country?

4.      The introduction is not acceptable in this form. It should describe the problem, global trends of broccoli crop, the need for this study, the proposed hypothesis, and lastly the objectives of this review article. I suggest extending it to not less than 500 words.

5.      Line 331: our recent results? Even If this report is from the authors, still I suggest not using impersonal terms like we, us, our, etc.

6.      A diagram showing the trend of selected climatic variables is a must in order to understand the clear fluctuations in the study timeframe.

7.      Draw a diagram depicting various impacts of climate change on broccoli. It may include impact groups such as morphological, growth, biochemical, economic, etc.

8.      Correct the syntax errors present throughout the manuscript.

9.      How useful are the findings reported in this review for global broccoli production? The climatic trends may vary greatly in tropical regions too. Do authors suggest any measures for regions experiencing the highest climatic fluctuations? Also, any future plans for the least affected regions?

10.   Brassica oleracea L. is used too much. I suggest writing it as full at first use followed by B. oleracea. Follow international nomenclature rules for all botanical names.

Author Response

Please see the attached file Reviewer 1_Answer with our response to Reviewer 1's comments.

Reviewer 2 Report

This manuscript reviewed the temperature requirements in broccoli crop production and the potential impact of climate change on broccoli. There are many long sentences throughout the manuscript. Some sentences are incomplete and not clear. Please see examples below.

Line 72-79: this is a very long sentence. Consider rewriting.

Line 79-84: another long sentence. Consider rewriting.

Line 112: check the sentence “…is required in at least in some genotypes”.

Line 138: check the sentence “…can only be accurately identified only by microscop”.

Line 155-161: again, very long sentence.

Line l64: check the sentence “…this suggests the involvement, in addition to temperature and other factors”.

Line 443: “… resistance to enemies and diseases…”, => “…resistance to pests and diseases…”?

Line 455: check the sentence “…progress is often gradual, environmental variation large and heritability low.”

Author Response

Please see the attached file Reviewer 2_Answer with our response to Reviewer 2's comments.

Round 2

Reviewer 1 Report

The authors have adequately answered and revised the manuscript as per my comments. I did not see any reason hindering the publication of this paper. I suggest acceptance in current form. 

Author Response

We are grateful for the reviewer's constructive comments, which contributed to a better presentation of the manuscript.